# Residential Greenness, Lifestyle, and Vitamin D: A Longitudinal Cohort of South Asian Origin and Caucasian Ethnicity Women Living in the South of the UK

**DOI:** 10.3390/nu16081214

**Published:** 2024-04-19

**Authors:** Keila Valente de Souza de Santana, Helena Ribeiro, Andrea Darling, Israel Henrique Ribeiro Rios, Susan Lanham-New

**Affiliations:** 1Departamento de Saúde Ambiental, Faculdade de Saúde Pública, Universidade de São Paulo, São Paulo 01246-904, Brazil; lena@usp.br (H.R.); israelhenriquerr@usp.br (I.H.R.R.); 2Department of Nutritional Sciences, Faculty of Health and Medical Sciences, University of Surrey, Guildford GU2 7XH, UK; a.l.darling@surrey.ac.uk (A.D.); s.lanham-new@surrey.ac.uk (S.L.-N.)

**Keywords:** vitamin D, ultraviolet radiation, residential greenness, ethnicity, parathyroid hormone

## Abstract

The global population is at risk of vitamin D deficiency due to low exposure to sunlight and low intake of the vitamin through diet. The aim of this study was to investigate in women the association between vitamin D status and parathyroid hormone (PTH), ultraviolet radiation, lifestyle, ethnicity, social conditions, and residential greenness. A 1-year longitudinal study assessed vitamin D status in 309 women living at latitude 51°14′ N. Blood samples were taken four times throughout the year for analysis of 25(OH)D and serum PTH concentration. After each seasonal visit, the individuals completed 4-day diet diaries and used two dosimeter badges for 1 week to estimate weekly UVR exposure. A questionnaire was applied to provide information about lifestyle and their ethnicity. Residential greenness was measured by Normalized Difference Vegetation Index (NDVI), within a 1000 m radius around each participant’s home address. Women living in greener spaces were more likely to have improved vitamin D status (RR: 1.51; 95%CI: 1.13–2.02), as well as those who were more exposed to UVR (RR: 2.05; 95%CI: 1.44–2.92). Our results provide an insight into the connection between residential greenness, lifestyle, and vitamin D status comparing two ethnicities in a country with a temperate climate and with a high degree of urbanization.

## 1. Introduction

The synthesis of vitamin D in the epidermis is stimulated by solar ultraviolet B (UVB) radiation—between 280 and 315 nanometers (nm)—and is essential for human health, especially bone health. Vitamin D deficiency is recognized as a pandemic, and its major cause is the lack of appreciation that sun exposure in moderation is the major source of vitamin D for most humans [1,2]. Cultural aspects such as clothing that covers more of the body and the habit of spending more time indoors can be related to low vitamin D status [3,4,5,6].

Individual factors, such as sex, ethnicity, age, genetics, and physiology, can affect vitamin D status [3]. Studies also showed an inverse correlation between vitamin D levels and parathyroid hormone (PTH) [5]. PTH is a hormone produced and secreted by parathyroid cells. High levels of PTH cause an increase in the level of calcium in the blood [5].

Conversely, penetration of UVB photons in the human skin for the vitamin synthesis is affected by several environmental factors, such as season, time of the day, latitude, and lifestyle—for instance, use of sunscreen and cultural history [4,5,6]. There are still few studies that seek to discover the role of green exposure in residential areas in the synthesis of vitamin D [6,7,8]. Despite the recommendations of further studies, publications on the association between vitamin D levels and atmospheric and environmental factors remain limited.

Vitamin D deficiency is a widespread and common problem in the UK population, and is especially common in women and more severe in minority ethnic groups such as South Asian women, due to veiled clothing use for religious and cultural reasons, and consequently reduced sun exposure [4,5,9]. Thus, assessments of the individual and combined effects of residential greenness in the vicinity of homes, ultraviolet radiation (UVR), and lifestyle factors on vitamin D levels in women are necessary. A new approach to the vitamin D deficiency pandemic could involve nature-based solutions (NbS), which are considered climate change adaptation strategies [6]. Such strategies use ecosystem services to protect and sustainably manage natural or modified ecosystems, providing human well-being and biodiversity benefits [10]. However, the intimate relation between environmental conditions and human health is insufficiently explored within the NbS context. Trans-sectoral and trans-disciplinary efforts are required to improve the relative lack of literature on the possible relation between NbS and public health [11].

In this context, the study of vitamin D status from the perspective of NbS can bring new ways of understanding how the environment and lifestyle can contribute to health. Access to new technologies is essential for analyses of the association between residential greenness and the outcome of vitamin D deficiency, as they contribute to the inclusion of co-exposure factors. In addition, geotechnologies, such as remote sensing and geoprocessing, have contributed to epidemiological analyses that consider variables inherent to the environment and territory. The aim of this study was to investigate, in women, the association between vitamin D status and parathyroid hormone (PTH), ultraviolet radiation, lifestyle, ethnicity, social conditions, and residential greenness.

## 2. Materials and Methods

This research used data from the prospective cohort study D-FINES (Diet, Food Intake, Nutrition, and Exposure to the Sun in Southern England) conducted by the University of Surrey in Guildford, United Kingdom. The study is in accordance with the ethical standards established in the Declaration of Helsinki of 1964, the approval of the Research Ethics Committee (National Health Service NHS REC 06/Q1909/1 and University of Surrey EC/2006/19/SBMS) and the written consent of all participants were obtained. At the beginning of the study a questionnaire was applied, in which the participants provided information about lifestyle and their ethnicity, Caucasian or Asian.

Blood samples were collected at four time points in the year 2006-2007 for analysis of 25(OH)D and serum PTH concentration from all subjects: Summer (21 June 2006 to 20 September 2006), Autumn (21 September 2006 to 20 December 2006), Winter (21 December 2006 to 20 March 2007), and Spring (21 April 2007 to 20 June 2007). Two attempts were always made with collection of blood for both the Caucasian and Asian women and, where possible, by two different phlebotomists in the Clinical Investigations Unit at the University of Surrey.

After each seasonal visit, the individuals completed 4-day diet diaries and used two dosimeter badges for 1 week to estimate weekly UVR exposure (dosimetry details are found next).

### 2.1. Study Location

In this study, 25-hydroxyvitamin D [25(OH)D], PTH, UVR, and lifestyle variables were comparatively analyzed between two regions of southern England with diverse ethnicity, social conditions, and lifestyle: 

Region 1—Region farthest from south London consisting of Guildford, Farnham, and Basingstoke.

Region 2—Region closest to south London consisting of Croydon, Kingston, and Woking.

The analysis was also performed considering the two regions together.

### 2.2. Recruitments’ Method

A total of 365 subjects were recruited to the study at baseline (Summer 2006), with 279 Caucasian women and 86 Asian descendent women participating in the first measurements. By the end of the fourth visit, 223 Caucasian women and 70 Asian women attended the study on all four occasions. This study required at baseline 156 Caucasian women (78 premenopausal and 78 postmenopausal) for 80% power and 38 Asian women (19 premenopausal and 19 postmenopausal) for sufficient study power.

This study achieved 166 Caucasian women (98 postmenopausal and 68 premenopausal) and 41 Asian women (23 postmenopausal and 18 premenopausal) with four complete measurements of the key factors, particularly vitamin D status. The data on postmenopausal Caucasian women were previously published [12], as well as the analysis of premenopausal Caucasian and premenopausal South Asian women [3].

Power calculations for the Caucasian women were structured on the differences in 25(OH)D between women living in Southern England (Surrey) and women living in Northern UK (Aberdeen). To explore regional differences, a minimum of 78 Caucasian subjects would be needed at each site (Aberdeen and Surrey) to detect a difference of 9 nmol/L (equivalent to 0.4 SD) with 80% power [9]. Power calculations for the Asian women were structured on the differences in 25(OH)D between Caucasian and Asian women from published studies [9,12]. From each ethnic group, 26 subjects would be needed to detect a difference of 18 nmol/L (0.8 SD) with 80% power [12].

These calculations were very cautious given the longitudinal nature of the study, since the variance would be smaller and repeated measures/multi-level modeling could be used. The D-FINES team recruited subjects on the numbers for cross-sectional analysis. The particular strength of this study was the longitudinal nature of the study design and subsequent longitudinal analysis (repeated measures and multi-level modeling). It was not possible to calculate the subject numbers for the required study power when using longitudinal analysis, but certainly it would be less than that required for a single measurement.

The original hypothesis of the D-FINES study was not related to geographic investigation. Therefore, the ethnic division of the sample was not created in such a way that it was representative of individual geographic areas. Most of the selected Asian participants were from Croydon, Woking, and Kingston, as these areas had community groups that kindly assisted in the selection. Thus, the number of Asian people in the data, particularly for Region 2, is much higher than would be expected from the actual demographics of these areas of England.

A random sample of Caucasian women was recruited via the databases of general practitioner (GP) surgeries in Surrey, Hampshire, Berkshire, and outer London. The summer of 2006 was very hot (in relation to the usual summer weather in the United Kingdom) and Caucasian women probably had arms and legs partially bare (e.g., wearing short-sleeved t-shirt and shorts/skirt). In the fall, winter, and spring, Caucasian women probably covered their arms and legs due to the cold. English women of South Asian descent fulfilling the same inclusion criteria were recruited via local Asian Women’s networks. Seventy-five percent of Asian women were of Pakistani origin and wore veils that only showed their hands and face, having minimal skin exposure to the sun. Some of the Asian women were Hindu and wore a style of dress similar to that of Muslim women, but they did not wear a veil. Caucasian women had white skin and Asian women had darker skin pigmentation.

The recruitment method ensured sampling from a variety of areas of differing socioeconomic status that reflected the southeast of England in order to reduce risk of bias towards higher socioeconomic status participants.

### 2.3. Index of Multiple Deprivation (IMD)

Socioeconomic status was assessed by the Index of Multiple Deprivation (IMD). This was calculated from the Lower Super Output Area rank using the Office for National Statistics web site and was converted to the corresponding IMD using the Indices of Deprivation for Super Output Areas [13]. The 2006 IMD addressed seven domains related to income deprivation, job deprivation, health deprivation and disability, deprivation of education, skills and training, barriers to housing and services, deprivation of the living environment, and crime. The IMD is an aggregation at the level area-weighted of these specific dimensions of deprivation. The index was calculated based on the Lower Super Output Area classification using the Office for National Statistics website [13].

### 2.4. Evaluation of 25(OH)D and PTH Concentration

The laboratory used for the biochemical assay measurements (Vitamin D Research Group, Manchester Royal Infirmary, Manchester, UK) participates in the Vitamin D External Quality Assessment Scheme (DEQAS). The assay laboratory is accredited to ISO 9001:2015 [14] and ISO 13485:2016 [15] and participates successfully in the Vitamin D quality assurance scheme (DEQAS).

Serum 25(OH)D was measured using manual enzyme immunoassay (Immunodiagnostic Systems Ltd., Boldon, Tyne and Wear, UK). The manufacturer’s reference ranges were 19–58 ng/mL (48–144 nmol/L), but vary with season; sensitivity, 2 ng/mL (5 nmol/L); and intra- and inter-assay coefficients of variation, 6 and 7%, respectively (manufacturer values).

Serum intact PTH was measured using the OCTEIA enzyme-linked immunosorbent assay (Immunodiagnostic Systems Ltd., Boldon, Tyne and Wear, UK). The normal reference range for adults is 0.8 to 3.9 pmol/L; sensitivity, 0.06 pmol/L; and intra- and inter-assay coefficients of variation, 4 and 6%, respectively (manufacturer values).

### 2.5. Vitamin D Intake

To measure dietary vitamin D intake, four-day estimated food records (EPIC food diary) were used, which included one weekend day and three weekdays at each of the four time points in the year 2006–2007 [3].

The response of the Asian and Caucasian premenopausal women was lower than their postmenopausal counterparts and all efforts were made by the D-FINES team to keep returns of these diaries as high as was absolutely possible. Persuasion by the Asian networks was also particularly helpful. All women received a thank you letter on the return of their diaries or two reminder letters if the diaries were not returned. The D-FINES team could not have emphasized this further. All women received a detailed output of their dietary results.

### 2.6. UVR Dosimetry

UVR dosimetry was performed with badges made of polysulfone film that detects the amount of ultraviolet light and converts it into standard erythema dose (SED) units. Commonly used as a measure of UVR exposure, 1 SED is equivalent to 100 Jm^−2^ of erythemal UVR (sunburn), and is considered an acceptable daily dose of exposure [16].

For UV measurement purposes, the badges were read at 330 nm on the spectrophotometer (Aquarius, CECIL CE7200) at the University of Surrey (CV < 1%) before and after use. Participants wore one badge on their shoulder outer clothing on weekdays and the other badge on weekends, with their SEDs added together to provide an estimate of the weekly dose of SED. Participants filled out a sun exposure diary, but it was poorly filled out and, therefore, the data could not be used.

### 2.7. Normalized Difference Vegetation Index—NDVI

For the evaluation of the surrounding residential vegetation, the Normalized Difference Vegetation Index (NDVI) was calculated, defined as the amount of photosynthetically active vegetation and based on the reflectance of the visible spectrum of the Earth’s surface (red) and of the infrared close to the spectrum [17]. NDVI ranges from −1 to 1, values from 0.2 to 0.5 are associated with sparse and growing vegetation, and values above 0.5 indicate upper canopy of dense and healthy vegetation. Negative NDVI values correspond mainly to bodies of water [18]. However, in the study area, there were no large bodies of water, in such a way that the negative values could affect the final averages of greenness.

In the present study, NDVI is derived from cloudless Landsat 5 Thematic Mapper (TM) images, with spatial resolution of 30 m × 30 m (multispectral). The images available at the end of the spring season, June 2008, were obtained from the United States Geological Survey from the website https://earthexplorer.usgs.gov/, accessed on 20 February 2023.

The addresses of the participants’ residences were georeferenced according to the postal code, street name, and house number. For each participant, the surrounding residential greenness was abstracted as the NDVI average in buffers of 1000 m around their georeferenced address. The modification of the effect of residential greenness was investigated using a cut-off value above and below the median of the average NDVI in the 1000 m buffer [18].

### 2.8. Data Analysis

All statistical analyses were performed using R 3.6.1. Nonparametric Mann–Whitney U and T-Student tests were used for cross-sectional analyses. Inspection of the histograms for normality showed a skewed distribution for 25(OH)D, with the log transformation unable to normalize the data. Thus, a multilevel logistic regression model was used to evaluate the relative contribution of residential greenness, UVR exposure, vitamin D intake, and ethnicity to serum 25(OH)D levels. Data are presented as relative risk (95% confidence intervals).

## 3. Results

### 3.1. Nonparametric Results

This study sought to incorporate geographical variables of location and residential greenery in order to understand whether and how such variables could interfere or not with vitamin D status. In Table 1 we show the characteristics of the participants according to the region of residence to verify if there were significant differences between groups. Chi-square test showed a difference between regions 1 and 2 in relation to the ethnic group (*p* = 0.01). In addition, the Mann–Whitney U test showed a significant difference between the regions for IMD (*p* < 0.001) and residential greenness (Region 1 = 0.28 and Region 2 = 0.24, *p* < 0.001). There were no significant differences by ethnicity for age (*p* = 0.29).

In Table 2, the aim was to compare Caucasian women with Asian women, regarding accumulative frequencies throughout the year. In both groups, the percentage of participants with 25(OH)D < 25 nmol/L was lower in the summer, with 0.4% of Caucasian and 48.8% of Asian women. The proportion of Caucasian women with 25(OH)D < 25 nmol/L was higher in the winter, with 8.5%, while for Asian women it was higher in the fall, with 77.2%. The 50 nmol/L cutoff point was used for deficiency, as recently recommended by the new IOM guidelines [19]. Throughout the seasons, over 90% of the women in the Asian group remained with values of 25(OH)D < 50 nmol/L, while in the Caucasian group it ranged from 22.1% in the summer to 68.1% in the winter. 

In Table 3, the aim was to demonstrate the variation between ethnic groups in the different seasons of the year. Asian women had significantly lower 25(OH)D throughout the year. In both groups, the highest levels of 25(OH)D were in the summer, decreasing in the fall, reaching the lowest level in the winter, increasing again in the spring of 2007 and with values similar to the spring of 2008. According to the Mann–Whitney analysis, the variation in all seasons was significant in Region 1 and Region 2, as well as in the joint analysis of both regions. UVR exposure on outer clothing was also significantly different among ethnic groups in the summer, both in the joint analysis and in the individual analysis of each region. In the other seasons, although statistically significant in some cases, there was no difference in the absolute values of UVR exposure. Asian women had higher PTH levels throughout the year, being statistically significant in the summer in all analyses, Regions 1 and 2, and in the joint analysis. The index of residential greenness was higher in the Caucasian group than in the Asian group, in Region 1 (*p* = 0.03) and in the joint analysis of the regions (*p* = 0.03).

In Table 4, the aim was to show the total walking time per day by ethnic group and region, throughout the year. It tended to be higher in the Caucasian group than in the Asian group. Although, in the joint analysis of the regions, there was no significant difference between the groups in the winter (*p* = 0.08) and, in Region 1, in the summer (0.12) and fall (*p* = 0.06), Caucasian women demonstrated to have constancy in walking time, while the time spent by Asian women in the activity varied throughout the seasons. Sleep time tended to be slightly longer in the Caucasian group than in the Asian group in the summer and spring, especially in the joint analysis of the regions (*p* = 0.004 and *p* = 0.002, respectively).

In Table 5, the aim was to show data on hours a day and days a week in housework and paid work by ethnic group and region, throughout the year. According to the analysis of the data, Asian women spent more hours a day and days a week on housework in all seasons. The significant difference between the groups was within each region, mainly together and in all seasons (*p* < 0.001). The Caucasian group, in turn, showed to have more hours per day and days per week in paid work by region and together, and the difference between the groups, throughout the year, was also significant in its majority. 

The analysis of data on hours per day in moderate and intense functional activity showed no significant difference between the groups, within each region, and together, throughout the year. Although the hours per day in moderate and intense functional activity showed significant difference between the groups in the joint analysis of the regions, in the fall season (*p* = 0.01), and in Region 1, in the spring (*p* = 0.002), there was no difference in terms of absolute values. The same result was verified for the number of times per week that the activity causes shortness of breath while working/not working, with a significant difference between the groups in the analysis of the regions, but no difference in terms of absolute values.

### 3.2. Multi-Level Logistic Regression Model

Table 6 shows the multilevel model constructed by adding the predictor variables dietary vitamin D intake, exposure to UV rays, residential greenness, PTH, and ethnicity. No significant difference was found between the groups of premenopausal and postmenopausal women, with the latter group consisting only of women over 50 years of age. In addition, Body Mass Index (BMI) did not show a significant difference between the Caucasian and Asian groups and was not a significant parameter to the multilevel model to explain vitamin D [3]. 

The risk of vitamin D deficiency was higher in the Asian group (RR: 2.4; 95%CI: 1.4–3.4). The participants who had PTH levels above the median had higher risk of vitamin D deficiency (RR: 1.3; 95%CI: 0.4–2.1). The parameters residential greenness, UVR, and vitamin D intake were not significant and, therefore, did not improve the model’s prediction. 

It is important to highlight that the diet diaries showed that vitamin D intake stayed the same throughout the year but was higher in the Caucasian group than in the Asian group [3]. In both groups, half of the vitamin D consumed was from meat and fish, and around a fifth came from cereal products.

## 4. Discussion

The seasonal variation in 25(OH)D concentration in Caucasian women was as expected, with the highest levels in the summer, decreasing in the fall, reaching the minimum in the winter, and increasing again in the spring. In contrast, the Asian population showed seasonal constancy in 25(OH)D status, with over 90% of women having 25(OH)D concentrations below 50 nmol/L. Moreover, PTH levels of Asian women were higher than those of the Caucasian population in all seasons, with vitamin D status associated with the hormone [20]. In all seasons and in all regions of analysis, the significantly smaller difference of 25(OH)D for Asian women raises the need for formulating health policies to combat vitamin D deficiency for this ethnic group.

Sun-avoidance behavior among Asians has been described in observational studies on Middle Eastern women [21,22]. The lower sun exposure recorded in the Asian group by the dosimeters, in relation to the Caucasian group, suggests that the Asian women stayed outdoors for less time. The total time walking per day (in minutes) and hours per day and days per week in paid work was significantly lower in the Asian group in almost all study regions and seasons. In contrast, the number of hours per day and days per week in housework was higher for Asian women, demonstrating that the ethnic group may tend to stay indoors longer. The sleep quality and time of the Asian group must also be better evaluated, as in the present study there was a tendency for it to be lower than in the Caucasian group. 

This study area has a well-consolidated urban structure, with no difference in NDVI values between the studied regions. Studies show that the beneficial effects of residential greenness are more evident and significant in areas with a high level of urbanization and in more deprived neighborhoods, pointing to their significant potential to promote activities in urban environments [23]. Residential greenness has contributed to health promotion with a protective effect against many diseases, such as obesity and hypertension, favoring the reduction of all-cause mortality [23,24,25]. The benefit of residential greenness for vitamin D synthesis has followed previous findings, according to which it would be protective against deficiency [1,7,8]. More research evidence on the relationship between residential greenness, physical activity, UVR, and vitamin D status is necessary to establish the exact functional causality. However, considering the association between UVR exposure, vitamin D status, and varied health outcomes, more studies are necessary to discuss the possibility of vitamin D being a biomarker of health promotion in residential greenness [7,8].

Studies on residential greenness and health have shown that NDVI influences health behavior and outcomes, as it acts as a metric that captures the intangible salutogenic potential of the residential environment [25]. A study analysis by NDVI has also contributed to the production of evidence showing that nature plays a critical role in meeting social needs, which include the issue of health and well-being, as proposed by NbS approaches [10]. In addition to the estimation of exposure to residential greenness by average NDVI, a favorable point of the study was the measurement of other factors that influenced individual outdoor behavior. To date, there have been few studies on the relations between the functional role of residential greenness as environmental spaces conducive to sun exposure and vitamin D synthesis [1,7,8].

The wearing of clothing that covers more of the skin of Asian women is probably one of the causes of their lower concentrations of 25(OH)D, as clothes significantly reduce the formation of vitamin D in the skin [3,26]. The combination of sun avoidance, veiling, and also darker skin pigment is likely to be responsible for the lower 25(OH)D and a lack of seasonal change in 25(OH)D in the Asian than in the Caucasian women [3]. More research is needed to understand optimal parameters for green allocation and design for it to act as an upstream-level public health intervention ameliorating negative health externalities [25]. The planning of residential greenness should consider spatial population density profiles as well as their residential, demographic, and cultural characteristics, as such characteristics influence access and appropriation of public spaces.

In this study, we found no significant difference between the ethnic groups, for most of the regions of analysis, in the number of hours of moderate and intense physical activity. Furthermore, although a significant difference was found in relation to inactivity between the groups, the median and interquartile range values were quite similar. The analysis of such variables together with residential greenness can bring new ways to measure their protective effect resulting from their functional role for higher levels of vitamin D. Studies that address health promotion find beneficial effects of green urban spaces on walking, physical activity, and thermal comfort [7,25]. Such factors may contribute to greater exposure to sunlight, and influence vitamin D levels. 

Studies focusing on natural experiments may be feasible by analyses of individuals moving to new addresses with significant differences in green exposure, thus associating changes in vitamin D status with changes in green exposure before and after such changes. However, selection bias, especially in temperate countries, should be considered amid the possibility that participants with chronic diseases, such as obesity or even severe vitamin D deficiency, have selectively migrated to greener spaces, leading to the underestimation of the effects of residential greenness on the prevalence of deficiency.

To better carry out mediation analysis and elucidate the mechanisms by which residential greenness contributes to higher vitamin D levels, further studies based on epidemiological analysis methodologies, such as validated questionnaires on time spent outdoors during a sunny day, are necessary. Data obtained from technologies to stimulate healthy lifestyle and instruments for measuring physical activity, such as a smart watch, can bring more accurate individual information about causal pathways, despite the social selection.

Our study has some limitations. This study was limited to looking at vitamin D status in women, but given that deficiency is a common problem across the UK population, further study is needed for both sexes. Genetic influences have a role to play in 25(OH)D differences, but they were not evaluated in the present study. The lower concentration and lack of seasonal change of 25(OH)D in Asian women may be influenced by skin pigmentation. There was no assessment of the influence of clothing on the capacity of sun exposure on vitamin D synthesis in the Asian population. The use of self-reported data on physical activity and walking time is also subject to individual biases. Differences regarding the ethnic origin of the participants and the socioeconomic status of the geographic regions analyzed do not have population representativeness, which has become another limiting factor. In addition, we did not analyze air pollution, as studies suggest that air pollutants reduce the effectiveness of sun exposure in the production of vitamin D in the skin, absorbing and spreading solar radiation [7,27]. 

Similar to most observational studies, the present study measured residential greenness at an aggregate level of analysis using satellite-derived metrics of lower spatial resolution. Despite the limitation in the accuracy and generalization of data, the NDVI employed in the study proved to be an accessible measure of exposure of the density and quality of the green space. Moreover, the possible environmental changes resulting from climate change should be considered in the analysis of vitamin D status, as they will affect thermal comfort conditions, the quality of residential greenness, and air pollution, sharpening vitamin D deficiency [6]. 

This study contributed to a better understanding of how green areas can influence sun exposure behavior and, consequently, vitamin D levels. The strengths of this study include the longitudinal study design and detailed covariates at the individual level as well as confounding factors. Our results provided an insight into the connection between vitamin D, residential greenness, and lifestyle that may be important for urban and social planning to encourage outdoor activities in a region of temperate climate and with a high degree of urbanization.

## 5. Conclusions

This study deepened the knowledge of the relationship between climate, UVR, green areas, lifestyle, ethnicity, PTH, and vitamin D status. By addressing changes in the concentrations of 25(OH)D, serum PTH, and average UVR exposure throughout the seasons, together with ethnic factors—Asian and Caucasian—and geographic factors—location and green areas—not only were statistically significant differences observed, but also a noticeable variation in values between the different groups. The same aspects were observed in variables related to lifestyle, with emphasis on walking time, in which a constancy in walking time was noted in the Caucasian group, while in the Asian group there was relevant variation throughout the seasons of the year. 

However, this study also pointed out that despite the statistically significant difference in residential greenery between the groups, this does not reflect a real difference in exposure to greenery, given that NDVI values would need to have a more significant variation. Data modeling demonstrated that ethnicity and PTH were the variables that significantly interfered with vitamin D status, probably due to factors of less sun exposure in the Asian ethnic group because of the culture of greater body coverage and variables related to lifestyle that led this group to fewer outdoor activities. For this ethnic group, exposure to the sun should be even more encouraged, and more accessible green spaces can be an important factor that will favor daily outdoor recreation.

Although the dataset was studied from the period 2006-2007, analyses regarding data on the lifestyle of Caucasian and Asian women had not yet been carried out. Studies on the association between lifestyle and vitamin D status are still few, and there is a need to explore the topic further. Nowadays, when multidisciplinary approaches are sought to address health issues, the inclusion of residential greenness as an analysis variable related to the natural environment and to climate is a need in the field of nutrition. In addition, to our knowledge, this is the first study that analyzed the association between residential vegetation, lifestyle, and vitamin D status comparing two ethnicities in a country with a temperate climate.

## Figures and Tables

**Table 1 nutrients-16-01214-t001:** Baseline characteristics of pre- and postmenopausal Caucasian (*n* = 271) and Asian (*n* = 94) women by region in southeastern England.

		Region 1	Region 2	*p*-Value
Ethnicity	Asian	15 (16%)	79 (84%)	0.01
	Caucasian	162 (60%)	109 (40%)	
Age		55 (35–61)	49 (36.8–59.2)	0.29
IMD		14.1 (11.3–15.3)	18.8 (5.8–30.8)	<0.001
Residential greenness		0.28 (0.24–0.3)	0.24 (0.21–0.28)	<0.001

**Table 2 nutrients-16-01214-t002:** Cumulative frequencies (in percentage) by commonly used serum 25(0H)D status thresholds (in nanomoles per liter) for a cohort of pre- and postmenopausal women living in southeastern England.

Region 1 + 2			
25(OH) D	<25	<50	<75
Summer			
Caucasian	0.4	22.1	68
Asian	48.8	<91.7	<98.8
Fall			
Caucasian	1.9	41.4	80
Asian	77.2	<96.5	100
Winter			
Caucasian	8.5	68.1	<94.7
Asian	71.7	95	98.3
Spring			
Caucasian	7.1	<61.7	<91.8
Asian	70.2	93	100

**Table 3 nutrients-16-01214-t003:** Serum 25(OH)D, serum PTH, UV exposure, and residential greenness per ethnicity, region, and season for women living in southeastern England.

	Season	Region	Caucasian	Asian	*p*-Value
Serum 25(OH)D (nmol/L)	Summer	1	66 (52.4–82.1)	35.4 (22.6–52.8)	<0.001
		2	64.5 (50.8–77.6)	25 (20.6–30.4)	<0.001
		1 + 2	65.1 (51.7–79.5)	25 (20.4–32.1)	<0.001
	Fall	1	54.2 (44–73.8)	22.7 (15.2–46.2)	0.002
		2	54 (41–67)	19.1 (15.3–22.9)	<0.001
		1 + 2	53.9 (43.5–58.3)	18.7 (14.9–24.5)	<0.001
	Winter	1	43.7 ± 16.5	27.4 ± 14	<0.001
		2	41 (30.7–54.1)	16.2 (13.7–23.7)	<0.001
		1 + 2	40.7 (30.3–53.3)	16.7 (13.8–25.3)	<0.001
	Spring 2007	1	49.6 ± 23.1	28.8 ± 15.4	<0.001
		2	46.8 ± 17	23.5 ± 13.3	<0.001
		1 + 2	44.3 (34.3–57.7)	19 (14.7–30.6)	<0.001
	Spring 2008	1	48.3 ± 16.2	36.1 ± 15.5	0.02
		2	49 ± 21.2	25.9 ± 11.7	<0.001
		1 + 2	45.9 (35.9–63.6)	23 (18.3–32.5)	<0.001
Average UVR exposure (SED)	Summer	1	3.6 (1.8–6.04)	0.7 (0.4–1.7)	<0.001
		2	2.6 (1.5–5.1)	0.8 (0.1–1.4)	<0.001
		1 + 2	3.2 (1.6–5.9)	0.7 (0.2–1.5)	<0.001
	Fall	1	0.11 (0.01–0.6)	0.12 (0.01–0.3)	0.16
		2	0.2 (0.03–0.7)	0.06 (0.02–0.2)	<0.001
		1 + 2	0.2 (0.03–0.7)	0.1 (0.01–0.2)	0.003
	Winter	1	0.1 (0.05–0.3)	0.1 (0.07–0.2)	0.35
		2	0.1 (0.1–0.2)	0.1 (0.04–0.3)	0.71
		1 + 2	0.1 (0.06–0.3)	0.1 (0.06–0.3)	0.48
	Spring	1	2.4 (1.2–3.4)	1.6 (1.1–2.6)	0.23
		2	1.4 (0.6–2.6)	1 (0.3–2.4)	0.13
		1 + 2	1.9 (0.8–3.2)	1 (0.4–2.4)	0.007
PTH (pmol/L)	Summer	1	2.8 (2.1–3.5)	3.45 (2.85–3.95)	0.02
		2	2.6 ± 1.1	3.5 ± 1.6	<0.001
		1 + 2	2.6 (2–3.4)	3.4 (2.7–4.3)	<0.001
	Fall	1	2.9 ± 1.2	3.4 ± 1.2	0.1
		2	2.6 ± 1.1	3.9 ± 1.5	<0.001
		1 + 2	2.8 ± 1.1	3.8 ± 1.5	<0.001
	Winter	1	2.9 ± 1.2	3.2 ± 1.2	0.24
		2	2.7 ± 1.1	4.2 ± 1.8	<0.001
		1 + 2	2.8 ± 1.2	4.1 ± 1.7	<0.001
	Spring	1	2.9 ± 1.2	3.5 ± 1.6	0.1
		2	2.5 ± 1	3.9 ± 2	<0.001
		1 + 2	2.7 ± 1.1	3.8 ± 1.9	<0.001
Residential greenness	Spring	1	0.28 (0.24–0.3)	0.29 (0.28–0.31)	0.03
		2	0.23 (0.21–0.28)	0.23 (0.21–0.28)	0.21
		1 + 2	0.27 (0.21–0.29)	0.24 (0.22–0.27)	0.03

Values: mean ± SD or median (percentile 250–750). Statistical analysis: Mann–Whitney U, unless otherwise noted; independent *t*-test. Region 1—region farthest from south London consisting of Guildford, Farnham, and Basingstoke. Region 2—region closest to south London consisting of Croydon, Kingston, and Woking.

**Table 4 nutrients-16-01214-t004:** Total minutes of walking per day and hours spent in bed on workday/nonwork day by ethnicity, season, and region for women living in southeastern England.

Activity	Season	Region	Caucasian	Asian	*p*-Value
Total time walking per day (in minutes)	Summer	1	60 (28–120)	42.5 (11–60)	0.12
		2	60 (30–120)	60 (20–90)	0.02
		1 + 2	60 (30–120)	55 (20–90)	0.02
	Fall	1	60 (30–120)	40 (12–60)	0.06
		2	60 (30–120)	30 (0–90)	<0.001
		1 + 2	60 (30–120)	30 (0–82.5)	<0.001
	Winter	1	60 (20–120)	60 (15–90)	0.65
		2	60 (30–120)	30 (5–75)	0.01
		1 + 2	60 (30–120)	40 (15–90)	0.08
	Spring	1	60 (30–120)	15 (0–30)	0.02
		2	60 (30–120)	45 (20–81)	0.04
		1 + 2	60 (30–120)	45 (15–70)	0.005

Statistical analysis: Mann–Whitney U, median (250–750 percentile). Region 1—region farthest from south London consisting of Guildford, Farnham, and Basingstoke. Region 2—region closest to south London consisting of Croydon, Kingston, and Woking.

**Table 5 nutrients-16-01214-t005:** Hours a day and days a week in housework, hours a day and days a week in paid work by ethnicity, season, and region for women living in southeastern England.

Activity	Season	Region	Caucasian	Asian	*p*-Value
Hours per day in housework	Summer	1	2 (1–2)	3 (2–3.8)	0.005
		2	1 (1–2)	3 (2–4)	<0.001
		1 + 2	1.5 (1–2)	3 (2–4)	<0.001
	Fall	1	1 (1–2)	3 (1.1–4)	0.06
		2	1 (1–2)	3 (2–4)	<0.001
		1 + 2	1 (1–2)	3 (1.6–4)	<0.001
	Winter	1	1 (1–2)	3 (1.5–4)	0.04
		2	1 (1–2)	3 (1–4)	<0.001
		1 + 2	1 (1–2)	2.5 (1–4)	<0.001
	Spring	1	1 (1–2)	2 (1–4)	0.03
		2	1 (1–2)	2.5 (1.3–4)	<0.001
		1 + 2	1 (1–2)	2 (1–4)	<0.001
Hours per day in paid work	Summer	1	5 (0–8)	0 (0–7.9)	0.13
		2	6 (0–8)	0 (0–5.3)	<0.001
		1 + 2	5 (0–8)	0 (0–5.3)	<0.001
	Fall	1	5 (0–7.8)	0 (0–7.5)	0.13
		2	6 (0–8)	0 (0–7.5)	<0.001
		1 + 2	5 (0–8)	0 (0–7.5)	<0.001
	Winter	1	4.5 (0–7.5)	0 (0–0)	0.04
		2	6 (0–8)	2 (0–6)	0.002
		1 + 2	5 (0–8)	0 (0–5.3)	<0.001
	Spring	1	5 (0–7.8)	0 (0–7.5)	0.13
		2	6 (0–8)	0 (0–7.5)	<0.001
		1 + 2	5 (0–8)	0 (0–7.5)	<0.001
Days per week in housework	Summer	1	4 (2–7)	7 (5.5–7)	<0.001
		2	4 (2–7)	7 (5–7)	<0.001
		1 + 2	4 (2–7)	7 (5–7)	<0.001
	Fall	1	4 (2–6.5)	6 (2–7)	0.31
		2	4 (2–7)	7 (5–7)	<0.001
		1 + 2	4 (2–7)	7 (5–7)	<0.001
	Winter	1	4 (2–7)	7 (2–7)	0.25
		2	5 (3–7)	7 (5–7)	0.002
		1 + 2	5 (2.8–7)	7 (4.8–7)	0.001
	Spring	1	5 (3–7)	7 (7–7)	0.01
		2	5 (3–6)	7 (5–7)	<0.001
		1 + 2	5 (3–7)	7 (6–7)	<0.001
Days per week in paid work	Summer	1	3 (0–5)	0 (0–4.3)	0.045
		2	3 (0–5)	0 (0–5)	0.009
		1 + 2	3 (0–5)	0 (0–5)	<0.001
	Fall	1	3 (0–5)	0 (0–5)	0.009
		2	3 (0–5)	0 (0–5)	0.01
		1 + 2	3 (0–5)	0 (0–5)	0.004
	Winter	1	3 (0–5)	0 (0–0)	0.03
		2	3 (0–5)	1 (0–5)	0.048
		1 + 2	3 (0–5)	0 (0–5)	0.01
	Spring	1	3 (0–5)	0 (0–5)	0.1
		2	3 (0–5)	0 (0–5)	0.03
		1 + 2	3 (0–5)	0 (0–5)	0.01

Statistical analysis: Mann–Whitney U, median (250–750 percentile). Region 1—region farthest from south London consisting of Guildford, Farnham, and Basingstoke. Region 2—region closest to south London consisting of Croydon, Kingston, and Woking.

**Table 6 nutrients-16-01214-t006:** Summary of regression parameters for the multilevel model explaining vitamin D deficiency.

Parameter	Estimate (Beta)	LCL (0.025)	UCL (0.975)	*p* Value
Intercept	−1.92	−2.54	−1.30	<0.0001
NDVI 1000	0.29	−0.34	0.92	0.37
UVR	0.52	−0.15	1.19	0.13
Vitamin D intake	0.46	−0.17	1.09	0.15
Ethnic Asian	2.38	1.37	3.38	<0.001
PTH	1.28	0.44	2.11	0.003

## Data Availability

The datasets presented in this article are not readily available due to technical considerations. Requests to access the datasets should be directed to Susan Lanham-New.

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
