# Peer review of "Residential Greenness, Lifestyle, and Vitamin D: A Longitudinal Cohort of South Asian Origin and Caucasian Ethnicity Women Living in the South of the UK"

_nutrients, 2024, doi:10.3390/nu16081214_

Round 1

Reviewer 1 Report

Comments and Suggestions for Authors

The study suggests a longitudinal studies of South Asian woman and Caucasian woman in UK - however, it seems that there's an imbalance in the study population of 41 vs 166 - how does the author justify this differences? It is suggested later that it is due to area differences but this imbalance would affect the analysis - so how was this controlled exactly?

The authors also claimed that they have ran a power analysis - what kind? Please elaborate with more details.

I've also noticed that the dataset is more than 20 years old. Where the first blood draw was done in 2006? How can the authors justify that their result is still novel and valid.

I'm unsure how the manuscript language is being written here - it sounded rather non-inclusive? on line 120 for example the authors called caucasian woman population in their study white women? Or that the authors need to point out that on line 126 that caucasian women had white skin and asian women had darker skin pigmentation? This just do not ring a correct tone.

I'd suggest the authors list all of their statistical approaches and it's goal in a different section, the manuscript is quite hard to navigate.

Table 1. I'm missing the point of why the authors have ran a statistical on the number of people, age, IMD, and greenness..?

So what exactly did Table 6 take into account? What was the x and y-s of the regression model and what factors are controlled?

Table 7. What is the 'adjusted' in the model?

Comments on the Quality of English Language

English reads fine

Author Response

Dear reviewer

Thank you for your review. We sent you the point-by-point response.

The study suggests a longitudinal studies of South Asian woman and Caucasian woman in UK - however, it seems that there's an imbalance in the study population of 41 vs 166 - how does the author justify this differences? It is suggested later that it is due to area differences but this imbalance would affect the analysis - so how was this controlled exactly?

The authors also claimed that they have ran a power analysis - what kind? Please elaborate with more details.

R. More details are shown in lines 116 to 131.

I've also noticed that the dataset is more than 20 years old. Where the first blood draw was done in 2006? How can the authors justify that their result is still novel and valid.

R. Although the dataset was studied from the period 2006-2007, analyzes regarding data on the lifestyle of Caucasian and Asian women had not yet been carried out. Studies on the association between lifestyle and vitamin D status are still few, and there is a need to explore the topic further. Especially at this time, when multidisciplinary approaches are sought to address health issues, the inclusion of a new analysis variable that is residential greenness sought to include themes related to the natural environment and climate that need to be discussed in the field of nutrition. In addition, to our knowledge, this is the first study that aimed to analyze the association between residential greenness and vitamin D status in a country with a temperate climate.

The same text is shown on lines 472 to 480.

I'm unsure how the manuscript language is being written here - it sounded rather non-inclusive? on line 120 for example the authors called caucasian woman population in their study white women? Or that the authors need to point out that on line 126 that caucasian women had white skin and asian women had darker skin pigmentation? This just do not ring a correct tone.

R. We change the text on lines 142, 143 and 149.

I'd suggest the authors list all of their statistical approaches and it's goal in a different section, the manuscript is quite hard to navigate.

R. Statistical approaches have been placed in different sections, in lines 236 to 323.

Table 1. I'm missing the point of why the authors have ran a statistical on the number of people, age, IMD, and greenness..?

R. The study sought to incorporate geographical variables of location and residential greenery in order to understand whether and how such variables could interfere or not with vitamin D status. Therefore, in Table 1 we show the characteristics of the participants according to the region of residence and whether there was significant differences between groups. 

The same text is on line 237 to 240.

So what exactly did Table 6 take into account? What was the x and y-s of the regression model and what factors are controlled?

Table 7. What is the 'adjusted' in the model?

R. These table 6 and 7 were deleted. A new table was added with the multi-level model on line 354 and a new explanation is in line 326 to 340.

Reviewer 2 Report

Comments and Suggestions for Authors

The manuscript entitled "Residential greenness, lifestyle and vitamin D: A Longitudinal 2 Cohort of South Asian origin and Caucasian ethnicity women living in the South of the UK" presents the results of an extensive statistical analysis regarding the influence of various environmental and lifestyle factors on serum 25-OH vitamin D levels in a specific population from the UK. Although the study is very interesting and used an innovative approach to examine sunlight exposure and vitamin D status, there are still some important concerns to be addressed.

First, the Introduction section should contain a brief motivation for the selection of women for this specific study. The authors highlighted that individual factors such as sex have a significant influence on vitamin D status. However, there is no specific explanation why men have been ruled out.

The same limitation should also be acknowledged in the Discussion as a limitation of this study.

Second, the Discussion section is too extensive and a final summary of the findings is missing. The authors should consider inserting a Conclusion section or summarizing the most important findings in the final paragraph of the manuscript. The final paragraph of the actual version is too general and makes it difficult for the reader to understand the main points of the research.

Furthermore, the authors should give some explanation on not introducing age, as a significant contributing factor to vitamin D status, in the statistical analysis. In addition, a crucial factor, the body mass index, has not been explored. Considering the differences observed between Caucasian and Asian women regarding total time walking per day, one can deduce that there should be a difference in body composition also.

The vitamin D intake was mentioned as a contributing factor to 25-OH vitamin D serum levels, however, it was not described how it was evaluated. It would be of great interest to know how this parameter was assessed and whether a dose adjustment should have been introduced or not. 

Minor issues:
The authors should consider making a timeline of the samplings for serum 25OH-vitamin D measurements in order to be easier to understand for the readers.
In Table 3 "SD" should be replaced with ± symbol.
Line 397-399 - references are missing.

Author Response

Dear reviewer

Thank you for your review. We sent you the point-by-point response.

First, the Introduction section should contain a brief motivation for the selection of women for this specific study. The authors highlighted that individual factors such as sex have a significant influence on vitamin D status. However, there is no specific explanation why men have been ruled out.

R. The specific explanation why the selection of women was placed in lines 53 to 56.

The same limitation should also be acknowledged in the Discussion as a limitation of this study.

R. Limitation is acknowledged in lines 439 to 441.

Second, the Discussion section is too extensive and a final summary of the findings is missing. The authors should consider inserting a Conclusion section or summarizing the most important findings in the final paragraph of the manuscript. The final paragraph of the actual version is too general and makes it difficult for the reader to understand the main points of the research.

R. We inserted a Conclusion section in lines 467 to 494.

Furthermore, the authors should give some explanation on not introducing age, as a significant contributing factor to vitamin D status, in the statistical analysis. In addition, a crucial factor, the body mass index, has not been explored. Considering the differences observed between Caucasian and Asian women regarding total time walking per day, one can deduce that there should be a difference in body composition also.

R. The explanation was placed in lines 346 to 351.

The vitamin D intake was mentioned as a contributing factor to 25-OH vitamin D serum levels, however, it was not described how it was evaluated. It would be of great interest to know how this parameter was assessed and whether a dose adjustment should have been introduced or not. 

R. The explanation was placed in lines 186 to 196.

Minor issues:
The authors should consider making a timeline of the samplings for serum 25OH-vitamin D measurements in order to be easier to understand for the readers.

R. The explanation was placed in line 83 to 88.

In Table 3 "SD" should be replaced with ± symbol.

R. It was replaced.

Line 397-399 - references are missing.

R. The paragraph was deleted because it was unrelated to our results.

Round 2

Reviewer 1 Report

Comments and Suggestions for Authors

I'd like to thank the reviewer for their responses.

Power analysis is clear and satisfactory.

The effect of residential greenness is already studied for example here in 2020: https://www.jamda.com/article/S1525-8610(20)30356-X/fulltext

2022: https://www.sciencedirect.com/science/article/pii/S0160412022004500

2023: https://pubmed.ncbi.nlm.nih.gov/36990343/

All done in China with temperate climate, so I'm still lacking justification on the novelty of this study. Unless it is the novelty of comparing the two ethnicities?

The explanation for the analysis is now satisfactory

Comments on the Quality of English Language

Minor typos exist, for example Section 2.2, 'recruitment' - please check accordingly 

Author Response

Dear reviewer,

Thank you for your review. We included the novelty of comparing the two ethnicities (lines 26 to 28 and 477 to 480).  We also reviewed the Minor typos.

Many thanks

Reviewer 2 Report

Comments and Suggestions for Authors

The authors responded to all the concerns raised by the reviewer and modified the manuscript accordingly, except one minor comment aiming Table 3, which was only partially modified.

Author Response

Dear reviewer,

Thank you for your review. Sorry for our mistake. We have completely modified the table 3 now.

Many thanks